# Liver Function—How to Screen and to Diagnose: Insights from Personal Experiences, Controlled Clinical Studies and Future Perspectives

**DOI:** 10.3390/jpm12101657

**Published:** 2022-10-05

**Authors:** Matthias Buechter, Guido Gerken

**Affiliations:** 1Department of Gastroenterology and Hepatology, Elisabeth Hospital, 58638 Iserlohn, Germany; 2Department of Gastroenterology and Hepatology, University Hospital Essen, University of Duisburg-Essen, 45147 Essen, Germany

**Keywords:** liver disease, liver function, LiMAx, elastography

## Abstract

Acute and chronic liver disease is a relevant problem worldwide. Liver function plays a crucial role in the course of liver diseases not only in estimating prognosis but also with regard to therapeutic interventions. Within this review, we discuss and evaluate different tools from screening to diagnosis and give insights from personal experiences, controlled clinical studies and future perspectives. Finally, we offer our novel diagnostic algorithm to screen patients with presumptive acute or chronic liver disease in the daily clinical routine.

Acute and chronic liver diseases are major causes of morbidity and mortality worldwide. Nowadays, liver cirrhosis is the most common non-neoplastic cause of death in Europe and the U.S. among diseases of the gastrointestinal tract [1,2,3]. From a pathophysiological point of view, acute or chronic inflammation leads to destruction and progressive fibrosis of the liver parenchyma. Liver cirrhosis represents the endpoint of this frequently creeping silent process lasting for years or even decades [4]. Ongoing liver damage with an increase in portal pressure leads to the development of portal hypertension (PH), which is a crucial factor in the history of cirrhosis. When portal-hypertension-induced hepatic decompensation events such as ascites, variceal bleeding, hepatic encephalopathy or hepatocellular carcinoma occur, mortality increases rapidly [5,6,7]. Acute-on-chronic liver failure (ACLF), defined as acute decompensation of chronic liver disease associated with (multiple) organ failures, is probably the most serious acute decompensation of cirrhosis with reported mortality rates of up to 80%, despite new insights in pathophysiology and optimal treatment [8,9,10].

Fortunately, progress in diagnostics and therapy over the last decades has improved the prognosis of patients suffering from chronic liver disease (CLD). Nowadays, a variety of different treatment modalities have become available depending upon the etiology and stage of CLD. However, the management of patients with CLD becomes more and more challenging since there is a clear focus on individualized medicine taking into account not only the patient’s underlying liver disease and its severity but also co-morbidities and disease-modifying factors.

When establishing treatment strategies for primary hepatocellular carcinoma (HCC), for example, we have to include important pathogenetic aspects into our personalized decision-making process. On the one hand, there is the tumor burden of the patient. On the other hand, there is the host with all individual co-morbidities and risk factors modifying the course and progression of the tumor disease. In addition, the molecular and immunological interaction between host and tumor is very important. Finally, different local and systemic targeted HCC treatment options have to be evaluated carefully not only regarding their effect on the tumor itself but also on their influence to downregulate liver function and liver capacity [11,12,13].

However, the liver itself is one of the most important human solid organs with multiple functions. Thus, liver cells cover not only metabolic homeostasis but also metabolism, production of nutrients such as carbohydrates, fats, proteins, vitamins, distribution, storage and synthesis of proteins and final metabolic products including drugs and toxins invading the liver cell cycle.

The first and important step in the evaluation of liver function is the complete and proper clinical examination of the patient. Patients with liver cirrhosis may have numerous physical examination findings that reflect the severity of the underlying liver disease. Although some symptoms and signs related to advanced CLD are nonspecific (e.g., abdominal pain, nausea, malaise), some findings are more characteristic and indicate complications of liver disease [14]. Key physical findings in patients with cirrhosis are, for example, hepatomegaly, splenomegaly, dilated abdominal veins, spider nevi and palmar erythema, whereas ascites, gastrointestinal bleeding, jaundice and encephalopathy point to hepatic decompensation and advanced portal hypertension with significantly impaired liver function [6,15,16]. In addition, severe muscle depletion (sarcopenia) resulting from an imbalance between protein synthesis and breakdown in patients with advanced stages of CLD is related to increased complications [17,18,19]. Moreover, a recently published study showed that body fat composition determines outcomes in patients with cirrhosis, while low subcutaneous fat levels were associated with a higher rate of cirrhosis-associated complications and mortality [20]. 

The next practical clinical approach to determine the stage and severity of liver disease covers different diagnostic tools using conventional blood parameters or so-called “serum biomarkers” such as cellular enzymes, albumin, bilirubin and coagulation proteins, as well as simple scoring systems based on these parameters. 

Then, different imaging techniques are usually applied to visualize the hepatic parenchyma, its surface and perfusion. Essentially, ultrasound is the first-line approach which is often combined with a high-resolution computed tomography (CT) scan and/or magnetic resonance imaging (MRI). CT scans and MRI play an important role in the objective assessment of cirrhosis and its complications (e.g., focal liver lesions, HCC, tumor staging, portal vein thrombosis, ascites). However, due to its limited availability (MRI), radiation exposure (CT scan) and poor diagnostic accuracy in less advanced stages of fibrosis, these techniques are not well suited as screening or monitoring tools in chronic liver disease [21,22,23]. 

Finally, immunological, biological and genetic markers should be taken into consideration. Although non-invasive techniques have been extensively examined with good evidence, liver histology obtained either by percutaneous or mini-laparoscopy-guided biopsy still plays an important role in the assessment of acute or chronic liver diseases, especially at first diagnosis and in uncertain situations [24,25,26]. Still, this invasive approach offers the unique possibility of direct tissue analysis with different techniques such as immunohistochemistry and molecular pathology. However, due to its invasiveness and risk of complications (especially bleeding), liver biopsy is unsuitable for screening and monitoring patients suffering from CLD, particularly when repeated interventions are required [27,28]. Furthermore, the gained histological specimen only represents 1/50,000 of the whole liver tissue and might therefore not reflect the true degree of inflammation, fibrosis or cirrhosis, despite an adequate sample size in diffuse parenchymal liver diseases. In addition to the problem of “intraobserver variation”, this so-called “sampling error” may lead to over- or underestimation of the real extent and severity of CLD with relevant consequences [29,30,31].

The enzymatic machinery in liver cell cytoplasm plays an important role in the evaluation of liver disease as it represents the basis for determination of liver function containing, on one hand, parameters of cholestasis, e.g., bilirubin, alkaline phosphatase and gamma-glutamyltransferase, and on the other hand, transaminases as inflammation markers, e.g., aspartate aminotransferase (AST), alanine aminotransferase (ALT) and glutamate dehydrogenase (GLDH), as well as synthesis parameters such as albumin, cholinesterase and coagulation factors, e.g., factors I, II, V, VII, VIII and X, respectively. Determination of liver transaminases (e.g., AST, ALT) is easy to perform, and they are therefore well suited as screening parameters. Although temporary elevation of these enzymes is not a harmful problem, higher and repeatedly elevated levels directly correlate with increased mortality [32,33,34]. Hence, it is important that family doctors focus on that. Usually, an increase in ALT levels is compatible with hepatocellular damage (“hepatocellular pattern”), while an increase in cholestasis markers such as alkaline phosphatase represents a cholestatic liver disease (“cholestatic pattern”) [35]. A mixed pattern of these markers can serve as a hint to diseases with both aspects. In addition, determination of the exact levels of serum transaminases can help to classify the activity, severity and stage of the present liver disease. For example, acute and fulminant viral hepatitis is accompanied by very high transaminase levels, whereas chronic active liver disease comes along with mild elevation of these enzymes. Interestingly, in patients with advanced stages of liver disease including liver cirrhosis, liver enzymes are not infrequently only slightly elevated or even within normal ranges. Overall, none of these enzymes or molecules summarized under the term “serum biomarkers” are of use by themselves but are useful when combined with each other or certain clinical parameters in marker panels or mathematical scoring systems in order to assess the extent of CLD. The so-called “non-invasive fibrosis scores” such as the AST-to-ALT ratio (AAR), AST-to-platelet ratio index (APRI), fibrotest, NAFLD fibrosis score (NFS) or fibrosis-4 (FIB-4) score have been established in recent years for clinical use and can help to indirectly quantify the stage of liver fibrosis [21,36,37,38]. Besides the advantage of non-invasiveness, these scores are ubiquitously available and cost-effective. However, their prognostic value is not clear-cut, since they are useful in the exclusion of advanced fibrosis and cirrhosis (sensitivity 64-92%, specificity 38-75%, AUROC 0.74-0.88) but do not distinguish well early and intermediate stages of fibrosis [21,38,39,40,41,42,43]. In addition, the Child–Pugh Score (CPS), the Albumin–Bilirubin (ALBI) Score and the Model of End-Stage Liver Disease (MELD) Score can be used to evaluate actual liver function and estimate prognosis [44,45,46,47]. However, all these scoring systems have to be interpreted with caution since they include subjective evaluation (e.g., extent of ascites for the CPS) and harbor several pitfalls such as inflammation and malnutrition, which can influence relevant parameters such as albumin as an integral part of those scores. Furthermore, various laboratory data are not liver-specific (e.g., AST elevation due to alcohol abuse or muscle breakdown) and are biased by various factors such as deficiencies, (iatrogenic) substitution, drugs (e.g., downregulation of clotting factors by vitamin K antagonists), extrahepatic causes of, for example, thrombocytopenia or biological half-life of these enzymes. 

Conventional abdominal ultrasound (US) is unequivocally the most common and widely used imaging modality among patients with presumptive or established acute or chronic liver disease. This technique offers the opportunity to visualize hepatic parenchyma, its morphology and perfusion when combined with Doppler sonography. Characteristic findings of liver cirrhosis in US are nodular liver surface, round edge, inhomogeneous parenchyma with hypoechoic nodules, hypertrophy of the caudate segment and rarefication of liver veins [48,49]. The diagnostic accuracy in individuals with advanced cirrhosis is high (sensitivity and specificity > 90%), particularly when signs of (decompensated) portal hypertension (e.g., ascites, varices, splenomegaly) are present [50,51]. However, the accuracy of US in diagnosing fibrosis or (beginning) cirrhosis in the absence of portal hypertension is significantly inferior since liver morphology may be normal in these stages [21,52,53,54]. Although abdominal US is standardized in many countries (e.g., DEGUM certification in Germany), the quality of the examination highly depends on the skill, knowledge and experience of the investigator [55,56]. However, abdominal US does not only play a role in the initial diagnosis but is also very important in the surveillance of patients suffering from CLD: it is recommended as an integral part in the follow-up of cirrhotic patients in multiple international guidelines. In this regard, its fundamental importance lies in the early detection of complications such as the development of ascites, portal vein thrombosis or hepatocellular carcinoma (HCC), which strongly impacts patient management, outcome and economic burden of healthcare systems [57,58,59]. If a focal hepatic lesion in cirrhosis is detected by US, its characterization can be performed by injection of a contrast medium (contrast-enhanced ultrasound; CEUS). However, the role of CEUS in the detection and characterization of focal hepatic lesions is still discussed controversially, and its relevance is weighted differently in international guidelines. Nowadays, the combination of CEUS with high-resolution CT/MRI and determination of alpha-fetoprotein (AFP) is adequate to diagnose HCC (without the necessity of tissue acquisition by biopsy) [11,13,60,61,62]. 

During the last decades, ultrasound-based elastography has been introduced as a novel non-invasive imaging modality to stage liver fibrosis. By the use of transient elastography (TE) or shear-wave elastography (SWE) the stiffness (or hardness) of the liver tissue can be measured by low-frequency vibrations, which is supposed to be proportional to the extent of fibrosis [63,64,65,66]. Since progressive scarring of the liver parenchyma during the course of CLD leads to increasing elasticity, this technique can therefore not only aid in the initial characterization of disease severity but also serve as a tool for monitoring and treatment follow-up. Besides the advantage of non-invasivity, elastography is easy to perform, cost-effective and highly reproducible. There is reliable scientific evidence for this method strengthened by numerous studies, especially in the early and late stages of fibrosis (specificity and sensitivity > 90%) [67,68,69,70]. In advanced cirrhosis, liver and spleen elastography can provide additional information about the presence and extent of portal hypertension (PH), since they are positively correlated with the hepatic venous pressure gradient (HVPG) and decompensation events such as variceal bleeding [71,72]. However, anatomic conditions (e.g., body fat, presence of ascites, short rib distance) reduce its feasibility in approximately 10% of patients in whom valuable results cannot be attained. Furthermore, the results can be invalid or distorted by steatosis, congestion and inflammatory activity, particularly in terms of concomitant acute liver damage [36,73]. 

Decision making in clinical hepatology often requires the assessment of liver function. However, all the above-mentioned parameters are indirect markers for liver function and acute or chronic liver damage and therefore somehow more or less lack precision. Thus, it is sometimes difficult to assess the liver’s response to various insults, formulate a treatment approach and predict recovery by the use of these techniques [74]. However, how can we nowadays assess actual metabolic liver function more precisely?

Enzymatic liver function tests have been employed experimentally and clinically for several decades. What these tests have in common is that a metabolite of a usually intravenously applied substrate is measured in blood samples or exhaled air. The perfect substrate should be metabolized only in hepatocytes and therefore selectively reflect metabolic liver function. Different substrates of the cytochrome P-450 system such as ^13^C-aminopyrine, ^13^C-phenylalanine, ^13^C-galactose, ^13^C-methionine or monoethylglycinexylidide (MEGX) have been introduced and further investigated to estimate liver function during the last decades [74,75,76,77,78,79]. Moreover, indocyanine green (ICG) has been used to quantitatively assess liver function and hepatic clearance [80,81,82]. However, due to different reasons, these tests have not been established sustainably in the clinical routine. 

More recently, the dynamic measurement of the enzymatic liver function by the liver maximum capacity (LiMAx) breath test has been introduced as a robust technique to determine dynamic liver function based on the specific hepatic cytochrome p 450 1A 2 metabolism of an intravenously injected substance [83]. After application of the substrate _13_C methacetin, this is immediately demethylated into acetaminophen and _13_CO_2_ in hepatocytes. Then, the concentration of _13_CO_2_ is measured in exhalation, and the liver capacity can be calculated from the analysis of the _13_CO_2_/_12_CO_2_ ratio in relation to the individual baseline ratio prior to the substrate injection [84]. Thus, the procedure is visualized as a curve on the screen of a monitor. In healthy individuals, we usually can observe two parts of metabolic liver function. At first, there is a quickly rising curve representing the initial phase of the metabolizing process. After rapidly reaching the plateau (>500 μg/kg/h), usually after about 5 min, we see the long-term line representing the maintenance phase of liver function. If the liver function is significantly impaired (e.g., acute liver failure, advanced stages of cirrhosis), we observe completely different curve kinetics. After injection of the substrate, there is only one phase with a linear and very slowly rising gradient reaching the maximum not until the end of the examination after 60 min. In accordance, the maximum value is notably lower within this stadium (<100 μg/kg/h). On the other hand, in individuals with moderately impaired liver function (e.g., acute hepatitis, mild fibrosis), we see a sort of mixture of both curves. At first, there is a fast-rising curve similar to the healthy liver, although the gradient is less pronounced. After about 5 min, the gradient decreases to a slowly rising linear curve similar to the significantly impaired liver function but obviously being on a higher level. The three different curve kinetics of the LiMAx test are exemplarily demonstrated in Figure 1. In particular, we can therefore not only determine actual enzymatic function on the basis of the absolute and calculated liver maximum capacity value but also visualize, characterize and estimate dynamic hepatic reserve by the curve progression. 

We could confirm a good correlation between structural and functional changes in a cohort of 102 patients, in whom enzymatic liver function measured by LiMAx was closely associated with histologically proven parenchymal changes (fibrosis) and elastography determined by TE (Figure 2). However, the highest diagnostic accuracy of non-invasively detecting cirrhosis was reached by combining TE and LiMAx [85]. In addition, we found a strong correlation between different clinical stages occurring in the course of CLD and liver function in a great cohort of patients (n = 464) with CLD. Herein, the LiMAx test was even superior to TE, CPS, MELD and serum biomarkers with a Spearman’s correlation coefficient of −0.81 [86].

The LiMAx test offers the unique opportunity that alterations in liver function can be determined immediately and in real time. We here give an example of a young male patient with acute liver failure (ALF) of unknown origin with significantly impaired liver function complicated by sepsis, in whom the LiMAx test was the first parameter predicting beneficial outcome while laboratory parameters improved much later [87]. In particular, serial LiMAx measurements offer valuable additional information in the course of acute liver injury. We saw similar clinical courses in, for example, patients with ALF due to autoimmune hepatitis, where LiMAx was the first parameter estimating prognosis after initiation of high-dose steroids. These data were confirmed by two pilot studies in which LiMAx improved outcome prediction in ALF [88,89]. Likewise, estimating prognosis in CLD/cirrhosis is highly accurate by LiMAx and comparable to validated scoring systems such as the MELD [86,90]. The LiMAx test is therefore the first functional capacity test with added benefit to the current “standard of diagnostic care”. The findings of our group were confirmed by other tertiary European centers using the LiMAx test in Germany and the Netherlands [91,92,93,94,95].

Interventions in patients with CLD/liver cirrhosis may be associated with different complications. Further impairment of liver function, which can even lead to subsequent liver failure, is definitely the most important of them. Therefore, an adequate selection of patients suitable for a certain intervention is of fundamental importance. The LiMAx test was able to estimate prognosis with regard to liver surgery, transjugular intrahepatic portosystemic shunt (TIPS) or liver cancer therapies such as transarterial chemotherapy (TACE) or selective intra-arterial radiotherapy (SIRT; unpublished data) and decline from patients who will not benefit from these procedures [83,96,97,98,99,100,101]. However, it is not only possible to individually decide whether a certain intervention is possible but also to determine an appropriate timepoint. Chemotherapy-associated liver injury is a well-known phenomenon, and pre-operative chemotherapy is a major risk factor for postoperative liver failure. Herein, a study showed that LiMAx impairment was dependent on chemotherapy cycles and therapy-free intervals in patients receiving platin-based chemotherapy due to colorectal liver metastases. However, patients with an impaired LiMAx showed sufficient regeneration during chemotherapy cessation when surgery was postponed. Preoperative performance of the LiMAx test can therefore augment surgical strategy and timing of surgery after previous chemotherapy [102,103,104,105].

Artificial intelligence (AI) has come into the spotlight in medicine and gastroenterology. Hepatology is no exception, with a growing number of studies published that apply AI techniques to the diagnosis and treatment of liver diseases [106]. Deep learning models make it possible to extract clinically relevant information from diverse and complex clinical datasets. Imaging, laboratory data and histopathology, for example, contain information for detecting liver fibrosis, differentiating focal liver lesions and predicting the prognosis of chronic liver disease which AI can extract [107]. Ultimately, AI systems could be implemented in clinical routines as decision support tools [108].

To conclude, we have a bouquet of different diagnostic and prognostic tools to screen and estimate liver function in patients with acute and chronic liver diseases. These entities should be applied carefully and thoughtfully to our patients. With the LiMAx test, we have a strong and robust additional assay that offers the unique opportunity of specific, semiquantitative and dynamic measurement of actual enzymatic liver function, which we included in our novel diagnostic algorithm to screen patients with presumptive acute or chronic liver disease (Figure 3).

## Figures and Tables

**Figure 1 jpm-12-01657-f001:**
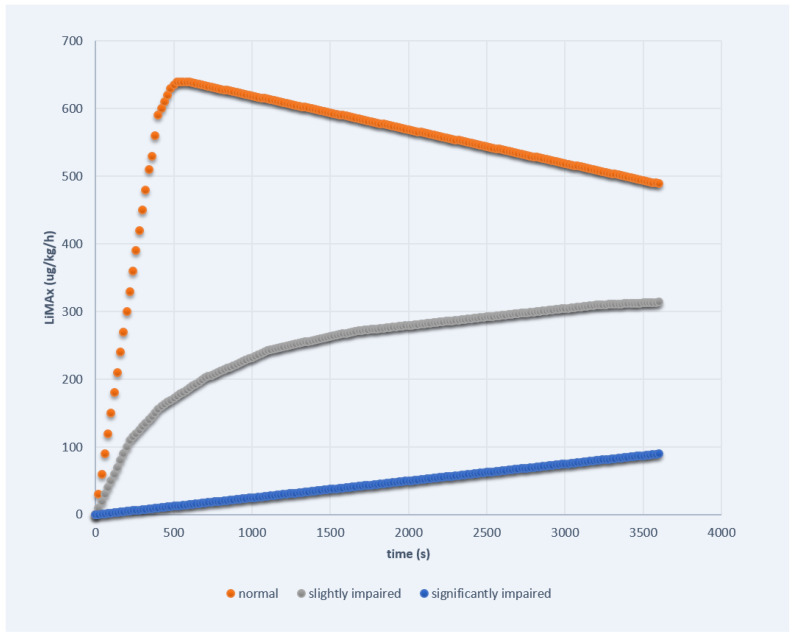
The three different curve kinetics of the LiMAx test.

**Figure 2 jpm-12-01657-f002:**
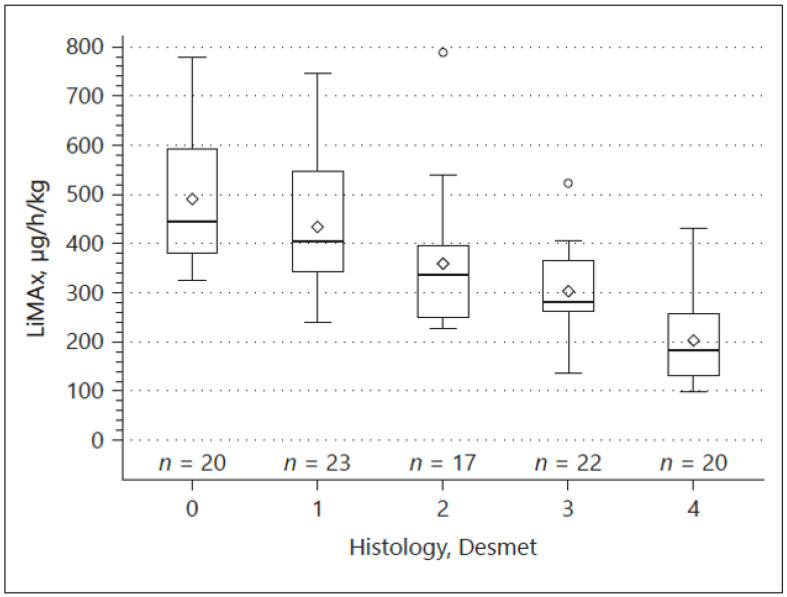
Correlation between LiMAx and histological specimen classified according to the Desmet scoring system.

**Figure 3 jpm-12-01657-f003:**
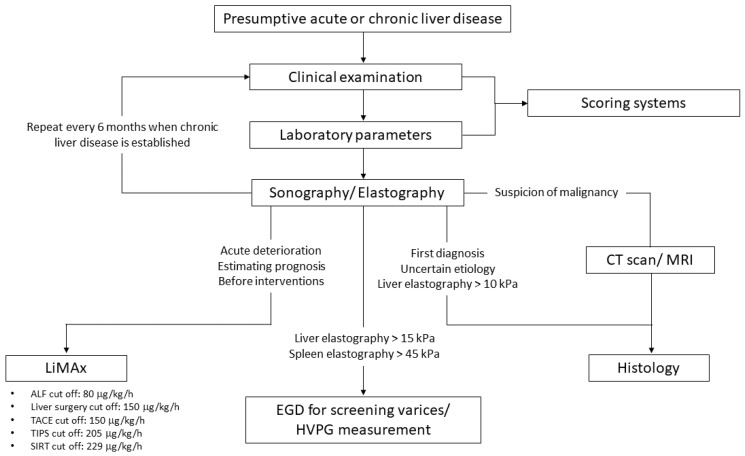
Proposed algorithm in individuals with presumptive acute or chronic liver disease.

## Data Availability

Data sharing not applicable.

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
