# Peer review of "Liver Function—How to Screen and to Diagnose: Insights from Personal Experiences, Controlled Clinical Studies and Future Perspectives"

_jpm, 2022, doi:10.3390/jpm12101657_

Round 1

Reviewer 1 Report

Authors have summarized the liver function test and given patchy algorithm

to this reviewer it seems an incomplete review it is more of a LiMax review rather than liver functions.

It is suggested to incorporate the all other tests and their relevance in non-invasive test

Reviewer 2 Report

Buechter et al.   discussed and evaluated different tools from screening to diagno[1]sis and give insights from personal experiences.The  review  is  well  written.  I would  advise  adding a  paragraph  about  the  studies focused  on artificial  intelligence  in  liver  diseases.  Thank  you  for  giving opportunty  to  review  this  study.

Reviewer 3 Report

Interesting paper!. 

I hope that the authors: 

1.     better comment physical examination, citing Wilson (2022)(1) . 

2.     briefly, but comprehensively, explain the importance of clinical scores, giving a brief explanation (with adequate citation of the literature) of the main ones and explaining their clinical usefulness

3.     analyze the importance of follow-up ultrasound in cirrhotic patients, recalling international guidelines and remembering its usefulness, including economic importance of early detection of complications (ascites, hepatocellular carcinoma, etc.)

4.     underline the usefulness of ultrasound with contrast media in the characterization of focal hepatic lesions in cirrhotic patients and the controversy in the use of this method at the level of international guidelines (American, European, Asian, Korean, Italian, etc.)

5.     Finally, in the final part of the reveiw, Authors would not only mention their own personal experience but also that of other centers, making the impact of LIMAX more scientific. 

in the attached pdf I have put some notes, where I suggest to insert an adequate reference to literature, or even propose some. This is for a greater completeness of the review

REFERENCES

1.         Wilson R, Williams DM. Cirrhosis. Med Clin North Am. 2022;106(3):437-46. 

Round 2

Reviewer 1 Report

Authors responded satisfactorily to the reviewer's comment

Reviewer 3 Report

Thank You for corrections. Good Job!!!